# Gender sensitivity and stereotypes in medical university students: An Italian cross-sectional study

**Fabrizio Bert[1], Edoardo Boietti[1]\*, Stefano Rousset[1], Erika Pompili[1], Eleonora Franzini Tibaldeo[1], Marta Gea[1], Giacomo Scaioli[1], Roberta Siliquini[1,2]**

**1** Department of Public Health Sciences, University of Turin, Turin, Italy, **2** AOU City of Health and Science, University of Turin, Turin, Italy

\* edoardo.boietti@unito.it

**Data Availability Statement:** Data contain potentially identifying or sensitive information, data sets are available upon request to the Corresponding Author and/or the Department of

## Abstract

Gender medicine is crucial to reduce health inequalities. Knowledge about students' attitudes and beliefs regarding men, women and gender is important to improve gender medicine courses. The aim of this study is to evaluate gender stereotypes and its predictors in Italian medical students. We performed an online cross-sectional study among students from the University of Turin. We used the validated Nijmegen Gender Awareness Scale in Medicine scale to explore gender sensitivity and stereotypes. Multivariable logistic regression model was performed to explore potential predictors of gender awareness. We enrolled 430 students. Female sex, a better knowledge on gender medicine and having had a tutor aware of gender issues are associated with higher gender sensitivity. Older age, a better knowledge on gender medicine and having had a tutor sensitive to gender issues were predictors of more stereotyped opinions towards patients. Having had a tutor aware of gender medicine, male sex and older age were associated with more stereotypes towards doctors. Italian students have high gender sensitivity and low gender stereotypes. Age, higher knowledge of gender medicine and having had a tutor that considered gender were associated with higher gender stereotypes. Focusing on gender awareness in medical schools can contribute to a better care.

## Introduction

Gender medicine is a transversal dimension of medicine, which describes within the same disease the differences of symptoms, clinical evolution, drug therapy and prevention between men and women. The goal of gender medicine is to understand the mechanisms through which gender differences influence health, the onset and course of many diseases and the outcomes of therapies [1–3]. For example, during clinical practice, gender of doctors and patients influences medical communication, patients' symptom presentations [4,5] and interpretations of patients' complaints and signs [6,7].

A Danish study with a sample of 6.9 million patients, reports that women receive a diagnosis of disease 4 years later than men [8]. Another study reports that the in-hospital mortality of

Public Health Sciences of University of Turin. This
Department is placed in Via Santena 5bis, Turin
Italy. Phone number:+39 011.6933864. E-mail:
direzione.dsspp@unito.it.

**Funding:** The authors received no specific funding
for this work.

**Competing interests:** The authors have declared
that no competing interests exist.

an acute myocardial infarction (AMI) is higher in women than in men up to 70 years of age
and survival after 6 months of AMI is lower in women [9].

In many diseases, such as coronary artery disease, Parkinson disease, irritable bowel syndrome, neck pain, knee joint arthrosis and tuberculosis, men are treated more extensively than
women [10]. Research found that physicians are more likely to consider symptoms reported
by men as organic [10]. These differences, if not properly considered, can lead to health
inequality. In particular, some pathologies can be undertreated in women compared to men or
vice versa. For this reason, it is important that doctors are aware of gender differences during
their clinical practice. The World Health Organization defines gender awareness as an 'understanding that there are socially determined differences between women and men based on
learned behaviour, which affect their ability to access and control resources' [11]. However,
medical phenomena are often both social and biological [12]. Gender awareness in doctors
aims to equity and equality in health. Nevertheless, in the past, gender was not considered in
medicine. Firstly, in the past medicine was 'gender blind'. Secondly, medicine seems to be
'male biased' because the largest body of knowledge on health and illness is about men and
their health. Thirdly, gender role ideology leads to wrong diagnosis and treatment. Gender
equity is not a spontaneous process. Gender medicine teaching may contribute to improve
gender knowledge. Implementation of gender medicine as a qualitative investment in medical
education is important for a future better health [13]. Gender awareness includes three components: gender-sensitivity, gender-role ideology and knowledge [14]. Gender sensitivity is the
'ability to perceive existing gender differences, issues and inequalities and incorporate these
into strategies and actions' [11]. Gender sensitivity includes the awareness that gender has an
impact on health, and affects the presentation of health complaints. This sensitivity allows
healthcare professionals to effectively address gender and improve care for both men and
women [15]. Gender-role ideology represents a health care worker's attitude towards male and
female patients and doctors [16]. Furthermore gender-role ideology is present at different levels in health services. Gender role ideology towards patients are a risk factor for inadequate
care. For example, several studies show that doctors often attribute psychological symptoms to
women for the same symptoms reported [17]. Gender role ideology towards doctors instead
refer to false beliefs such as thinking that female doctors are more empathic than male doctors
[18]. Gender awareness is important to prevent gender bias, to reduce health inequalities and
to improve 'patient-centered medicine'. Gender bias includes gender stereotype (unjustified
difference of treatment between female and male patients) and gender blindness (inability to
recognize differences when they are clinically pertinent).

Measuring medical students' attitudes and values concerning gender is possible using the
Nijmegen Gender Awareness Scale in Medicine, (N-GAMS). A Netherlands study shows that
male medical students held stronger gender stereotypes than female [1]. A Swedish study
found difference in gender sensitivity and gender -role ideology between Dutch and Swedish
student. Male students had more gender stereotypes than female. Age, father's birth country
and mother's education level had impact on gender sensitivity and stereotypes [19]. Another
study conducted in Portugal on medical students showed that empathy was associated with
higher gender sensitivity and lower endorsement of gender-role ideologies, while sexism was
associated to higher endorsement of gender-role ideologies and lower gender sensitivity [20].
In Switzerland, data from a observational study suggests a more gender stereotyped opinion
toward patients among male students; in addition, gender sensitivity increase while stereotypes
decreased with students getting older [21]. The focus on a medicine that takes gender differences into account begins in the 80s with the signing by the UN (United Nation) of a convention aimed at eliminating all forms of discrimination against women [22]. In Europe, a
growing interest in gender medicine led to the inclusion of this issue in the new Horizon 2020

research funding program [22]. In Italy, a *Plan for the application and diffusion of Gender Specific Medicine* (2019) aims to spread the principles of gender specific medicine that are not yet fully and adequately implemented in medical academic training [22]. For this reason, the aim of this study is to evaluate gender awareness through N-GAMS scale and its predictors in Italian male and female medical students, since no data exist in literature about gender awareness in our national context.

## Materials and methods

We performed an online cross-sectional survey among medical students of the University of Turin during March 2020. Students of the fifth and sixth years and outside prescribed time were considered eligible for this study. The questionnaire was submitted to students of the last years of the degree course because they already had practical training experiences with patients and had received an important part of theoretical training. All students were recruited and no one was excluded on the basis of gender, age or nationality. A sample of 1258 students received an email with the background, purpose of the study and the link to the online questionnaire. Informed consent was requested. Participation in the study was voluntary and anonymous. All procedures performed in studies involving human participants were in accordance with the ethical standards of the institutional and/or national research committee and with the 1964 Helsinki declaration and its later amendments or comparable ethical standards. The study was approved by the Internal Review Board of the Department of Public Health Sciences, University of Turin.

### The questionnaire

After a review of the literature, a 29-items questionnaire was developed, organized in five sections:

1. Socio-demographic characteristics of the sample (items 1–9);

2. Knowledge about gender medicine issues (items 10–13);

3. Attitudes of the students regarding gender medicine (items 14–19);

4. Gender sensitivity and stereotypes toward patients and doctors (item 20), using the validated questionnaire developed by Verdonk et al., the Nijmegen Gender Awareness In Medicine Scale (N-GAMS) [1];

5. Training experiences regarding gender medicine (items 21–29).

The questionnaire was in Italian-language. A bilingual researcher translate the tool from English to Italian. The translated N-GAMS scale was then tested for understandability in a pilot study recruiting 20 medical students.

This paper focuses on the fourth part of questionnaire, assessing gender sensitivity and gender stereotypes in medical students and their association with socio-demographic features, gender medicine knowledge and training experience regarding gender medicine.

Table 1 displays the variables assessed to describe the sample, stratified by gender. In particular, we explored the following characteristics: age, year of course, nationality, marital status, having children, self-perceived health status, having one or more chronic diseases, familiarity for chronic diseases. Self-perceived health status was assessed with a Likert scale, ranging from 1 (very bad) to 5 (very good), and it was subsequently dichotomized in "not good" (score 1 and 2) and "good/very good" (score 3, 4 and 5). Gender medicine knowledge was assessed with questions regarding the correct definitions of sex, gender and gender medicine, gender-related

**Table 1. Description of the sample (N = 430).**

| | | Females % (N) | Males % (N) | p-value |
|---|---|---|---|---|
| **Age** | *Mean±SD* | 25.20±2.26 | 25.11±1.86 | 0.689 |
| **Year of Course** | *Fifth* | 41.16 (121) | 46.67 (63) | 0.445 |
| | *Sixth (last)* | 31.29 (92) | 31.11 (42) | |
| | *Outside prescribed time* | 27.55 (81) | 22.22 (30) | |
| **Nationality** | *Italian* | 98.98 (291) | 99.26 (135) | 0.623 |
| | *Other* | 1.02 (3) | 0.74 (1) | |
| **Marital Status** | *Single* | 86.99 (254) | 93.38 (127) | **0.032** |
| | *With Partner* | 13.01 (38) | 6.62 (9) | |
| **Children** | *No* | 99.66 (293) | 100.00 (135) | 0.685 |
| | *Yes* | 0.34 (1) | 0.00 (0) | |
| **Health Status self-reported** | *Not good* | 14.29 (42) | 11.76 (16) | 0.291 |
| | *Good / Very Good* | 85.71 (252) | 88.24 (120) | |
| **Chronic Diseases** | *No* | 85.71 (252) | 88.97 (121) | 0.222 |
| | *Yes* | 14.29 (42) | 11.03 (15) | |
| **Familiarity for Chronic diseases** | *No* | 27.21 (80) | 27.94 (38) | 0.616 |
| | *Yes (parents / brothers /sisters)* | 68.37 (201) | 69.85 (95) | |
| | *Yes, other relatives* | 4.42 (13) | 2.21 (3) | |
| **Knowledge about Gender Medicine** | *Poor (under the mean of the sample)* | 41.84 (123) | 62.50 (85) | **<0.001** |
| | *Good (equal or above the mean of the sample)* | 58.16 (171) | 37.50 (51) | |
| **Have you ever dealt with gender medicine issues during lessons?** *Answer*: *Yes* | | 62.24 (183) | 71.32 (97) | **0.041** |
| **During traineeships in the wards, did you ever discuss with the tutor about the impact of sex and gender on patient management?** *Answer*: *Yes* | | 24.23 (71) | 41.04 855) | **<0.001** |
| **During the traineeships in the wards, did you have the impression that the tutor took into consideration sex and gender of patients during clinical practice?** *Answer*: *Yes* | | 40.61 (119) | 44.70 (59) | 0.247 |

epidemiology of frequent diseases, and true/false questions regarding specific gender medicine issues.

Personal experience during academic training were assessed asking the students whether they have dealt with gender medicine issues during lessons and during traineeships in the wards. To measure student's gender awareness, we used the N-GAMS scale. This scale explores two attitudinal aspects of gender-awareness: gender sensitivity (GS) and gender role ideology towards patients (GRIP) or doctors (GRID). We asked the students to self-report their agreement for each item, on a Likert scale ranging from 1 "not agree at all" to 5 "totally agree". Some sentences have reverse meaning; therefore, in these cases we calculated a reverse score. The GS score has 14 items, which investigate student's general opinion of considering gender and sex in healthcare; the GRIP score has 11 items, which explore the presence of stereotypes about male or female patients; the GRID score has 7 items, which investigate student's stereotypes towards practice of doctors. A higher GS score means a higher gender sensitivity. GRIP and GRID high score indicates more gender-stereotyping opinions. In this paper, concerning gender awareness and their predictors, we consider as outcome N-GAMS (the fourth part of the questionnaire).

## Statistical analysis

Descriptive analyses were carried out for all the variables. The continuous variable (age) was expressed as mean and standard deviation (SD). All the other variables were reported as percentages and numbers for each category. To evaluate differences between groups defined by the gender, Fisher's exact tests were calculated. Potential predictors of gender awareness were explored through a multivariable logistic regression model. The covariates to be included into

the model were selected using a stepwise forward selection process, with a univariate p<0.25 as the main criterion. Missing values were excluded by listwise deletion. All analyses were performed with the STATA 13 software, and a two-tailed p-value <0.05 was considered to be statistically significant.

## Results

### Socio-demographic characteristics

A sample of 430 students completed the questionnaire correctly. Females students were 294 (68.4%) and 136 (31.6%) were males. Sample mean age was 25.2±2.1. The 42.9% of the students attended fifth year of medical school, while the others were divided between sixth year and outside prescribed time. The knowledge about gender medicine is significantly different between male and female students (p<0.05).

The 41.8% of female students and 62.5% of male students had a poor knowledge of gender medicine (under the mean of the sample). There was a significant difference between males who reported having dealt gender medicine during lessons (71.3%) and females (62.2%) as well as during traineeships in the wards. Only about half of the sample had the impression that their tutor took into consideration sex and gender of patients during clinical practice. (Table 1).

### Stereotypes in gender medicine

A significant difference between male and female students was found with the GS sub-scale, with a mean score of 3.86±0.41 for female and 3.73±0.41 for male (p = 0.003). This suggests that female students had higher sensitivity to gender issues. As shown in Table 2, GRIP (the higher the score value the stronger the stereotypes toward patients) subscores were not significantly different between female and male students (1.80±0.57 for women, 1.87±0.65 for men). GRID subscore, instead, significantly differs between males (1.64±0.68) and females (1.51 ±0.49) which suggests that male students have significantly more stereotyped opinion toward doctors than females (p = 0.028) (Table 2).

### Potential predictors of GS, GRIP and GRID

We used a multivariable linear regression model to explore whether the socio-cultural background variables were related to outcome on GS, GRIP and GRID. Males had lower GS scores (coefB -0.96, CI95% -0.18 - -0.01, p-value = 0.030), while students who had a better knowledge on gender medicine (coefB 0.14, CI95% 0.0 6–0.22 p-value <0.001) and those who have received good example from the internship tutors (coefB 0.14, CI95% 0.06–0.26 p-value <0.001) were more gender-sensitive. (Table 3). Moreover, older students expressed more stereotypical thinking about patients (coefB 0.04, CI95% 0.01–0.07 p-value = 0.012). Surprisingly, students who had a better knowledge of gender medicine (coefB 0.12, CI95% 0.01–0.24 p-value = 0.040) and those who had the impression that their tutor took into consideration sex and gender of patients during clinical practice (coefB 0.13, CI95% 0.01–0.25 p-value = 0.045) agreed more with stereotypical thinking about patients (Table 4).

**Table 2. Stereotypes in gender medicine: Gender sensitivity (GS), role ideology toward patients (GRIP) and role ideology towards doctors (GRID).**

|  |  | Females | Males | p-value |
|---|---|---|---|---|
| **Gender Sensitivity (GS)** | *Mean±SD* | 3.86±0.41 | 3.73±0.41 | **0.003** |
| **Gender Role Ideology toward Patients (GRIP)** | *Mean±SD* | 1.80±0.57 | 1.87±0.65 | 0.263 |
| **Gender Role Ideology toward Doctors (GRID)** | *Mean±SD* | 1.51±0.49 | 1.64±0.68 | **0.028** |

**Table 3. Potential predictors of gender sensitivity.**

|  |  | Coef B | CI95% | p-value |
|---|---|---|---|---|
| **Age** |  | 0.01 | (-0.01–0.03) | 0.400 |
| **Sex** | *Female* | Ref | - | - |
|  | *Male* | -0.96 | (-0.18 - -0.01) | **0.030** |
| **Year of Course** | *Fifth* | Ref | - | - |
|  | *Sixth or more* | 0.02 | (-0.03–0.07) | 0.531 |
| **Marital Status** | *Single* | Ref | - | - |
|  | *With partner* | -0.07 | (-0.20–0.07) | 0.327 |
| **Health Status self-reported** | *Poor* | Ref | - | - |
|  | *Good / Very good* | -0.01 | (-0.13–0.11) | 0.875 |
| **Chronic Diseases** | *No* | Ref | - | - |
|  | *Yes* | -0.07 | (-0.19–0.05) | 0.232 |
| **Familiarity for Chronic diseases** | *No* | Ref | - | - |
|  | *Yes* | -0.01 | (-0.06–0.09) | 0.707 |
| **Knowledge about Gender Medicine** | *Poor* | Ref | - | - |
|  | *Good* | 0.14 | (0.06–0.22) | **<0.001** |
| **Have you ever dealt with gender medicine issues during lessons?** *Answer*: *Yes* | | -0.08 | (-0.16–0.00) | 0.050 |
| **During traineeships in the wards, did you ever discuss with the tutor about the impact of sex and gender on patient management?** *Answer*: *Yes* | | -0.01 | (-0.10–0.08) | 0.813 |
| **During the traineeships in the wards, did you have the impression that the tutor took into consideration sex and gender of patients during clinical practice?** *Answer*: *Yes* | | 0.14 | (0.06–0.26) | **<0.001** |

**Table 4. Potential predictors of gender role ideology towards patients (GRIP).**

|  |  | Coef B | CI95% | p-value |
|---|---|---|---|---|
| **Age** |  | 0.04 | (0.01–0.07) | **0.012** |
| **Sex** | *Female* | Ref | - | - |
|  | *Male* | 0.04 | (-0.08–0.17) | 0.490 |
| **Year of Course** | *Fifth* | Ref | - | - |
|  | *Sixth or more* | -0.07 | (-0.15–0.01) | 0.053 |
| **Marital Status** | *Single* | Ref | - | - |
|  | *With partner* | -0.04 | (-0.24–0.15) | 0.669 |
| **Health Status self-reported** | *Poor* | Ref | - | - |
|  | *Good / Very good* | 0.05 | (-0.13–0.22) | 0.593 |
| **Chronic Diseases** | *No* | Ref | - | - |
|  | *Yes* | 0.03 | (-0.14–0.20) | 0.740 |
| **Familiarity for Chronic diseases** | *No* | Ref | - | - |
|  | *Yes* | -0.07 | (-0.18–0.05) | 0.242 |
| **Knowledge about Gender Medicine** | *Poor* | Ref | - | - |
|  | *Good* | 0.12 | (0.01–0.24) | **0.040** |
| **Have you ever dealt with gender medicine issues during lessons?** *Answer*: *Yes* | | 0.12 | (0.01–0.24) | 0.050 |
| **During traineeships in the wards, did you ever discuss with the tutor about the impact of sex and gender on patient management?** *Answer*: *Yes* | | 0.06 | (-0.08–0.20) | 0.045 |
| **During the traineeships in the wards, did you have the impression that the tutor took into consideration sex and gender of patients during clinical practice?** *Answer*: *Yes* | | 0.13 | (0.01–0.25) | **0.045** |

Gender stereotypes towards doctors were higher in male students (coefB 0.12, CI95% 0.01–0.24 p-value = 0.046) and with increasing age (coefB 0.03, CI95% 0.01–0.06 p-value = 0.045). Having a tutor that took into consideration sex and gender of patients during clinical practice was associated with more stereotypical thinking about doctor (coefB 0.03, CI95% 0.01–0.06 p-value = 0.045) (Table 5).

## Discussion

To the best of our knowledge, there are no previous studies in Italy assessing Gender Awareness in medical students. The aim of this study was to evaluate gender awareness and to explore possible predictors of gender sensitivity and gender stereotypes towards patients and doctors in a sample of medical students of the University of Turin. Our students showed higher GS score compared to other European students. Probably, in Italy, there is a greater gender sensitivity because female social conquests are lower than northern Europe. With 63 out of 100 points, Italy ranks 14th in the EU on the Gender Equality Index [23] (12th in the domain of health). This probably increases student's consideration of gender issues [24]. We found higher Gender sensitivity in female students, probably because female suffer the consequences of gender inequalities. This gender difference is not found in similar studies conducted in Switzerland, Sweden and Netherlands [19,21] and it could be explained considering the specific socio-cultural context of each country. In fact, social status of women is better in many European countries compared to Italy. For example, in Sweden gender equality is highly considered in several social dimensions and in health-care services (Sweden ranks 1st in the EU on the Gender Equality Index) [19,24].

A better knowledge about gender medicine and having a tutor who took in consideration sex and gender of patients during clinical practice were associated with higher GS. Our results

**Table 5. Potential predictors of gender role ideology towards doctors (GRID).**

| | | Coef B | CI95% | p-value |
|---|---|---|---|---|
| **Age** | | 0.03 | (0.01–0.06) | **0.045** |
| **Sex** | *Female* | Ref | - | - |
| | *Male* | 0.12 | (0.01–0.24) | **0.046** |
| **Year of Course** | *Fifth* | Ref | - | - |
| | *Sixth or more* | -0.01 | (-0.08–0.06) | 0.771 |
| **Marital Status** | *Single* | Ref | - | - |
| | *With partner* | -0.11 | (-0.29–0.07) | 0.236 |
| **Health Status self-reported** | *Poor* | Ref | - | - |
| | *Good / Very good* | 0.02 | (-0.14–0.18) | 0.818 |
| **Chronic Diseases** | *No* | Ref | - | - |
| | *Yes* | -0.02 | (-0.17–0.14) | 0.850 |
| **Familiarity for Chronic diseases** | *No* | Ref | - | - |
| | *Yes* | -0.09 | (-0.20–0.01) | 0.082 |
| **Knowledge about Gender Medicine** | *Poor* | Ref | - | - |
| | *Good* | 0.11 | (-0.01–0.21) | 0.054 |
| **Have you ever dealt with gender medicine issues during lessons?** *Answer*: *Yes* | | 0.08 | (-0.03–0.19) | 0.182 |
| **During traineeships in the wards, did you ever discuss with the tutor about the impact of sex and gender on patient management?** *Answer*: *Yes* | | -0.02 | (-0.15–0.11) | 0.726 |
| **During the traineeships in the wards, did you have the impression that the tutor took into consideration sex and gender of patients during clinical practice?** *Answer*: *Yes* | | 0.15 | (0.03–0.26) | **0.014** |

suggest that the behaviour of the tutors influences student's gender sensitivity. We think that the tutors could stimulate interest towards this discipline. Other studies found an increasing GS score with age and lower score in students who had a father with different birth country [19,21]. Regarding gender stereotypes toward patients, we observed mean GRIP score of 1,87 in male and of 1,87. These results are comparable with a Swedish study [19] but they are lower than other results found in Netherlands and Switzerland [19,21]. Moreover we found that GRID score in our sample of Italian students is the lowest among European studies [19,21]. Despite living in a society where gender gap is still high (70th place in the global gender gap report 2019), Italian medical students have less stereotypical thinking regarding the role of gender in medicine compared to their colleagues in Europe [25]. It is important to consider that the European context is in any case much better in terms of gender inequalities in health than in developing countries [26]. Literature shows significant sex-related differences in gender-role ideologies towards patients in European studies, but in our study we observed sex-related difference in gender-role ideologies toward doctors (mean male GRID score is 1,64, mean female GRID score is 1,51; p = 0.028) but not toward patients. Consistently with our study, Portuguese data found less stereotypical thoughts toward doctor in females than males (21). Socio-cultural factors such as the Italian gender gap probably explain the highest scores in gender stereotypes in male students. Multivariable analysis shows an increasing GRIP score with age, better knowledge of gender medicine and having had a tutor that took into consideration gender and sex of patients during clinical practice. Previous studies show that stereotypes decreased with students getting older. Proceedings with their studies, Swedish and Swiss medical students reduce their stereotypes probably for a good theoretical and practical teaching system [19,21]. In Italy, the stereotypes observed were lower than other studies. Nonetheless, we found an increasing GRIP and GRID score with students' age. This is in contrast with other studies. The teaching of gender medicine and the student's experience during traineeship increase stereotypes and do not reduce it. Probably, students acquired the stereotypical gender difference as real gender differences. We found that a better knowledge and a good consideration of gender and sex of patients by the tutor were associated with higher gender stereotypes. A possible explanation of these results is a lack of awareness about stereotypes thoughts. A greater knowledge of real gender difference in medicine and greater interest of the tutor on gender and sex during clinical activities could lead to wrong acquisition of common gender beliefs. Nevertheless, our results indicate that GS is higher and stereotypes are lower than other European studies. Italian students are sensitive to gender medicine and have substantially few stereotypes. However, our country does not have a curricular program to prevent gender stereotypes toward both patients and doctors. It is important to teach the students that socio-cultural stereotypical differences are not real differences and are not part of gender medicine. Therefore, the implementation of gender specific teaching throughout elective courses should be seriously considered. Speaking about gender stereotypes in order to improve gender specific medicine is a priority. The teaching of gender medicine should reduce stereotypical thoughts and false beliefs and can contribute to create a real 'patient-centered medicine'.

## Limitations

The principal limitation of this study is that it was conducted on a convenient sample of students attending the last years of medical training; it could be interesting to know if gender awareness is different between freshman students and students of the last years. It is possible

that students previously sensitized or interested in the gender dimension of health answered the survey in a larger proportion and were over-represented.

## Strengths

We used a validated tool (N-GAMS) to assess gender awareness among medical students. This study is the first, to our knowledge, to have assessed gender awareness among Italian medical students. Other studies can be compared to our study in order to confirm our results in Italian medical student from different cities.

## Conclusions

In conclusion, the results of this study indicate that our students have high gender sensitivity and low gender stereotypes towards patients and doctors. However, age, higher knowledge of gender medicine and having had a tutor that took in consideration gender in clinical practice were associated with higher gender stereotypes. During gender courses and practical training more attention must be paid to explain that stereotyped gender differences are not scientifically proven and they do not contribute to provide a better care for both male and female patients. Further studies are needed for a better understanding of the factors that could reduce gender stereotypes in medical students.

## Acknowledgments

The Authors would like to thank all the students of the University of Turin who answered the questionnaire.

## Author Contributions

**Conceptualization:** Fabrizio Bert.

**Data curation:** Fabrizio Bert, Eleonora Franzini Tibaldeo.

**Formal analysis:** Fabrizio Bert.

**Investigation:** Fabrizio Bert, Erika Pompili, Eleonora Franzini Tibaldeo, Marta Gea.

**Methodology:** Fabrizio Bert, Stefano Rousset, Erika Pompili.

**Project administration:** Fabrizio Bert.

**Supervision:** Fabrizio Bert, Giacomo Scaioli, Roberta Siliquini.

**Validation:** Fabrizio Bert, Roberta Siliquini.

**Visualization:** Roberta Siliquini.

**Writing – original draft:** Edoardo Boietti, Stefano Rousset.

**Writing – review & editing:** Fabrizio Bert.

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
