## [Decision Letter · Decision Letter 0]

1 Sep 2021

PONE-D-21-25375

Are 2020’s medical students still suffering from gender stereotypes? An Italian cross-sectional study

PLOS ONE

Dear Dr. Boietti,

Thank you for submitting your manuscript to PLOS ONE. After careful consideration, we feel that it has merit but does not fully meet PLOS ONE’s publication criteria as it currently stands. Therefore, we invite you to submit a revised version of the manuscript that addresses the points raised during the review process.

We look forward to receiving your revised manuscript.

Kind regards,

Ramune Jacobsen

Academic Editor

PLOS ONE

Reviewers' comments:

Reviewer's Responses to Questions

**Comments to the Author**

1. Is the manuscript technically sound, and do the data support the conclusions?

Reviewer #1: Partly

Reviewer #2: Partly

Reviewer #3: Yes

Reviewer #4: Yes

Reviewer #5: Partly

2. Has the statistical analysis been performed appropriately and rigorously? 

Reviewer #1: Yes

Reviewer #2: No

Reviewer #3: Yes

Reviewer #4: Yes

Reviewer #5: Yes

3. Have the authors made all data underlying the findings in their manuscript fully available?

Reviewer #1: Yes

Reviewer #2: No

Reviewer #3: Yes

Reviewer #4: No

Reviewer #5: Yes

4. Is the manuscript presented in an intelligible fashion and written in standard English?

Reviewer #1: No

Reviewer #2: Yes

Reviewer #3: Yes

Reviewer #4: Yes

Reviewer #5: Yes

5. Review Comments to the Author

Reviewer #1: In many diseases, such as coronary artery disease, Parkinson disease, irritable bowel

syndrome, neck pain, knee joint arthrosis and tuberculosis, men are treated more extensively

than women (8).

Research found that physicians are more likely to consider symptoms reported by men as organic (8). Gender awareness in doctors aims to equity and equality in health

Method section

A sample of 1258 students received an email with

the background,

Reviewer #2: 1. The topic is relevant

2. I will suggest that the authors give a proper description of the variables that were used for measurement in the analysis.

3. Tables 1, 3, 4 and 5 in the results section have the same questions. This makes it difficult to ascertain the variables that are being measured and how the results were obtained. If possible, the authors should properly distinguish between the tables to make for easier understanding of the results.

4. I will also suggest that the authors provide a more detailed exploration of gender sensitivity and gender role ideology using, if possible, a theoretical framework.

5. The authors should include a flow chart to show how participants were recruited into the study.

6. The statement "With 63 out of 100 points, Italy ranks 14 in the EU on the Gender Equality Index....." in the Discussion section should be appropriately referenced.

7. The word "Globally" in the results section should be replaced with another word that describes the context of the study population.

Reviewer #3: Title and significance of the manuscript

The title of the manuscript succinctly and adequately describes the scope of the work reported in the article. The work is significant and worth publishing in PLOS ONE given the relevance and importance of the findings on gender stereotyping, be it for the practitioner or patient in medicine particularly as it concerns medical education and practice particularly in Italy and around the world in general.

The manuscript is well written in clear and simple language for the readers to understand.

Materials and methods

The data collection and analysis procedures were adequately presented by the authors. The description of the content of the study tool i.e. the questionnaire is good.

It is suggested that the ethical approval reference number assigned to the study protocol on which data this paper is based be provided in the last sentence of the first paragraph of the materials and methods section of the manuscript.

Results

It would be good if the authors could add the age range of the participants and their socio-demographic characteristics in Table 1.

In the results section, the titles of all the five tables presented in the paper ought to have been typed at the top of the tables contrary to the bottom of the tables as presently presented by the authors. This needs to be corrected.

Recommendation

The manuscript is publishable with minor corrections to be done by the authors.

Reviewer #4: This is an interesting study because gender profoundly influence interactions between health care providers and patients. Overall the manuscript represents valuable information regarding stereotypes in gender medicine including gender sensitivity (GS), role ideology toward patients (GRIP) and role ideology towards doctors (GRID). The study findings could be a good contribution to health science. Although there are important findings in the context of medical students in Italy, the design is repetition of previous work. In general, I recommend this paper to be considered for acceptance after responding several following comments:

1. Research regarding gender-based differential patterns of care by health providers arise predominantly from the developed countries. In my opinion, similar studies from developing countries are also important to be compared. Could you elaborate more on this issue? For example a study below presents the role that gender has on health.

Fikree, F. F., & Pasha, O. (2004). Role of gender in health disparity: the South Asian context. BMJ (Clinical research ed.), 328(7443), 823–826. https://doi.org/10.1136/bmj.328.7443.823

2. Page 5, the questionnaire section: After a review of the literature, a 29-items questionnaire was developed. Please be more detail on how to develop the questionnaire through the literature review.

3. This study used survey instrument- a questionnaire containing the N-GAMS scale to measure gender awareness. Is it in italian-language questionnaire? If yes, how the translation process and validation?

4. Kindly indicate the reasons why students of the fifth and sixth years and outside prescribed time were chosen for this study.

Reviewer #5: In many diseases, such as coronary artery disease, Parkinson disease, irritable bowel

syndrome, neck pain, knee joint arthrosis and tuberculosis, men are treated more extensively

than women (8). Research found that physicians are more likely to consider symptoms reported by men as organic (8).Gender awareness in doctors aims to equity and equality in health.

Did you mean to say these diseases are treated more extensively by men than women?

Kindly explain and paraphrase your sentence properly.

Sentence structure paraphrasing needed

6. PLOS authors have the option to publish the peer review history of their article (what does this mean?). If published, this will include your full peer review and any attached files.

Reviewer #1: No

Reviewer #2: No

Reviewer #3: No

Reviewer #4: No

Reviewer #5: No

---

## [Author Response · Author response to Decision Letter 0]

14 Oct 2021

Dear Editor, 

We are submitting the revised version of our manuscript “Are 2020’s medical students still suffering from gender stereotypes? An Italian cross-sectional study”, according to the comments of the reviewers.

Rev 1: 

We would like to thank you for your useful considerations.

Rev 2: 

1. I will suggest that the authors give a proper description of the variables that were used for measurement in the analysis.

Thank you for this useful suggestion. We added the following paragraph in the Materials and Methods section:

“This paper focuses on the fourth part of questionnaire, assessing gender sensitivity and gender stereotypes in medical students and their association with socio-demographic features, gender medicine knowledge and training experience regarding gender medicine.

Table 1 displays the variables assessed to describe the sample, stratified by gender. In particular, we explored the following characteristics: age, year of course, nationality, marital status, having children, self-perceived health status, having one or more chronic diseases, familiarity for chronic diseases. Self-perceived health status was assessed with a Likert scale, ranging from 1 (very bad) to 5 (very good), and it was subsequently dichotomized in “not good” (score 1 and 2) and “good/very good” (score 3, 4 and 5). Gender medicine knowledge was assessed with questions regarding the correct definitions of sex, gender and gender medicine, gender-related epidemiology of frequent diseases, and true/false questions regarding specific gender medicine issues. 

Personal experience during academic training were assessed asking the students whether they have dealt with gender medicine issues during lessons and during traineeships in the wards”.

2. Tables 1, 3, 4 and 5 in the results section have the same questions. This makes it difficult to ascertain the variables that are being measured and how the results were obtained. If possible, the authors should properly distinguish between the tables to make for easier understanding of the results.

Thank you for this suggestion. However, we think that a modification of the tables is not necessary, as they report the results of different analysis. In particular, table 1 reports the description of the sample, while tables 3, 4 and 5 report the results of the multivariable regression model exploring the potential predictors of Gender Sensitivity (table 3), GRIP (table 4) and GRID (table 5), as specified in the text.

3. I will also suggest that the authors provide a more detailed exploration of gender sensitivity and gender role ideology using, if possible, a theoretical framework.

Thank you for your comment. We modified as follow:

“Gender sensitivity includes the awareness that gender has an impact on health, and affects the presentation of health complaints. This sensitivity allows healthcare professionals to effectively address gender and improve care for both men and women (13). Gender-role ideology represents a health care worker’s attitude towards male and female patients and doctors (14). Furthermore gender-role ideology is present at different levels in health services. Gender role ideology towards patients are a risk factor for inadequate care. For example, several studies show that doctors often attribute psychological symptoms to women for the same symptoms reported (15). Gender role ideology towards doctors instead refer to false beliefs such as thinking that female doctors are more empathic than male doctors (16).”

4. The authors should include a flow chart to show how participants were recruited into the study.

Thank you for this comment. As described in the Materials and Methods section we sent the e-mail with the link to online questionnaire to all the medical students of the fourth, fifth year and outside prescribed times, with no exclusion criteria. A total of 1258 students received the e-mail and 430 of them filled in the questionnaire, thus representing our study population. We think that a flow chart would not add any other relevant information on the recruiting process. 

5. The statement "With 63 out of 100 points, Italy ranks 14 in the EU on the Gender Equality Index...." in the Discussion section should be appropriately referenced.

Thank you, we inserted the correct reference.

6. The word "Globally" in the results section should be replaced with another word that describes the context of the study population.

We replaced the word “Globally” with this sentence: “A sample of 430 students completed the questionnaire correctly.”

Rev 3

1. It is suggested that the ethical approval reference number assigned to the study protocol on which data this paper is based be provided in the last sentence of the first paragraph of the materials and methods section of the manuscript.

Thank you for your comment. We cannot provide a study protocol number because we used the Internal Review Board of the Department of Public Health Sciences, University of Turin.

2. It would be good if the authors could add the age range of the participants and their socio-demographic characteristics in Table 1.

Thank you for this remark. We inserted the missing socio-demographic features in Table 1 (Nationality and Children). Since the participants of this survey were all students with similar age (97% of the participants had an age between 22 and 30 years), we think that age should be properly expressed as mean and SD rather than categorizing it as age classes. 

3. In the results section, the titles of all the five tables presented in the paper ought to have been typed at the top of the tables contrary to the bottom of the tables as presently presented by the authors. This needs to be corrected.

We appreciated this comment and we corrected the titles of the tables as suggested.

Rev 4:

1. Research regarding gender-based differential patterns of care by health providers arise predominantly from the developed countries. In my opinion, similar studies from developing countries are also important to be compared. Could you elaborate more on this issue? For example a study below presents the role that gender has on health. Fikree, F. F., & Pasha, O. (2004). Role of gender in health disparity: the South Asian context. BMJ (Clinical research ed.), 328(7443), 823 826. https://doi.org/10.1136/bmj.328.7443.823

Thanks for your comment, we added this part in the discussion section: “It is important to consider that the European context is in any case much better in terms of gender inequalities in health than in developing countries (24).”

We think that it is difficult to compare our results with studies from developing countries. There are no data in the literature on gender awareness in medical students in developing countries.

2. Page 5, the questionnaire section: After a review of the literature, a 29-items questionnaire was developed. Please be more detail on how to develop the questionnaire through the literature review.

Thank you for this comment. As also requested by Reviewer n°2, we added the following paragraph describing the variables analyzed and the construction of the questionnaire. 

Table 1 displays the variables assessed to describe the sample, stratified by gender. In particular, we explored the following characteristics: age, year of course, nationality, marital status, having children, self-perceived health status, having one or more chronic diseases, familiarity for chronic diseases. Self-perceived health status was assessed with a Likert scale, ranging from 1 (very bad) to 5 (very good), and it was subsequently dichotomized in “not good” (score 1 and 2) and “good/very good” (score 3, 4 and 5). Gender medicine knowledge was assessed with questions regarding the correct definitions of sex, gender and gender medicine, gender-related epidemiology of frequent diseases, and true/false questions regarding specific gender medicine issues. 

Personal experience during academic training were assessed asking the students whether they have dealt with gender medicine issues during lessons and during traineeships in the wards”.

As specified in the text, the only part of the questionnaire derived from the literature was the N-GAMS scale, for which we provided the appropriate citation. 

3. This study used survey instrument- a questionnaire containing the N-GAMS scale to measure gender awareness. Is it in italian-language questionnaire? If yes, how the translation process and validation?

We appreciate this comment. We add this sentence: “The questionnaire was in Italian-language. A bilingual researcher translate the tool from English to Italian. The translated N-GAMS scale was then tested for understandability in a pilot study recruiting 20 medical students.”

4. Kindly indicate the reasons why students of the fifth and sixth years and outside prescribed time were chosen for this study.

We appreciate this comment. We completed the text in the Material and Methods section as follows: “The questionnaire was submitted to students of the last years of the degree course because they already had practical training experiences with patients and had received an important part of theoretical training”.

Rev 5: 

1. In many diseases, such as coronary artery disease, Parkinson disease, irritable bowel

syndrome, neck pain, knee joint arthrosis and tuberculosis, men are treated more extensively than women (8). Research found that physicians are more likely to consider symptoms reported by men as organic (8). Gender awareness in doctors aims to equity and equality in health. Did you mean to say these diseases are treated more extensively by men than women? Kindly explain and paraphrase your sentence properly.

Thank you for your consideration. We added information about this as follows:

In many diseases, such as coronary artery disease, Parkinson disease, irritable bowel syndrome, neck pain, knee joint arthrosis and tuberculosis, men are treated more extensively than women (8). Research found that physicians are more likely to consider symptoms reported by men as organic (8). These differences, if not properly considered, can lead to health inequality. In particular, some pathologies can be undertreated in women compared to men or vice versa. For this reason, it is important that doctors are aware of gender differences during their clinical practice.

---

## [Decision Letter · Decision Letter 1]

3 Nov 2021

PONE-D-21-25375R1Are 2020’s medical students still suffering from gender stereotypes? An Italian cross-sectional studyPLOS ONE

Dear Dr. Boietti,

Thank you for submitting your manuscript to PLOS ONE. After careful consideration, we feel that it has merit but does not fully meet PLOS ONE’s publication criteria as it currently stands. Therefore, we invite you to submit a revised version of the manuscript that addresses the points raised during the review process.

We look forward to receiving your revised manuscript.

Kind regards,

Ramune Jacobsen

Academic Editor

PLOS ONE

Journal Requirements:

Reviewers' comments:

Reviewer's Responses to Questions

**Comments to the Author**

1. If the authors have adequately addressed your comments raised in a previous round of review and you feel that this manuscript is now acceptable for publication, you may indicate that here to bypass the “Comments to the Author” section, enter your conflict of interest statement in the “Confidential to Editor” section, and submit your "Accept" recommendation.

Reviewer #1: All comments have been addressed

Reviewer #2: All comments have been addressed

Reviewer #3: All comments have been addressed

Reviewer #4: All comments have been addressed

Reviewer #5: All comments have been addressed

2. Is the manuscript technically sound, and do the data support the conclusions?

Reviewer #1: Partly

Reviewer #2: Yes

Reviewer #3: Yes

Reviewer #4: Yes

Reviewer #5: Partly

3. Has the statistical analysis been performed appropriately and rigorously? 

Reviewer #1: Yes

Reviewer #2: Yes

Reviewer #3: Yes

Reviewer #4: Yes

Reviewer #5: Yes

4. Have the authors made all data underlying the findings in their manuscript fully available?

Reviewer #1: Yes

Reviewer #2: Yes

Reviewer #3: Yes

Reviewer #4: Yes

Reviewer #5: Yes

5. Is the manuscript presented in an intelligible fashion and written in standard English?

Reviewer #1: Yes

Reviewer #2: Yes

Reviewer #3: Yes

Reviewer #4: Yes

Reviewer #5: Yes

6. Review Comments to the Author

Reviewer #1: 1 Title issue Very good article but my concern is that a research question cannot be a title.

Kindly consider and change the title into a statement and not a question, the question can be included as part of your introduction, along with your aim of study

4 Data absent Kindly include the data to reveal instances about gender differences influencing health.

Please state clearly the impact (state wise, nationally and globally) and effect on people’s health, physically, socially, mentally e.t.c

Which is one of the reasons why you must have chosen the topic.

Study Rationale Invariably including a strong rationale for this study would have brought out the best of this article

Reviewer #2: The authors have taken time to address all the comments that were raised by reviewers in the initial submission.

Reviewer #3: (No Response)

Reviewer #4: (No Response)

Reviewer #5: Please check the attached.

Tile, rationale and methods.

Pages

Queries

My suggestions to the author

1

Title issue

Very good article but my concern is that a research question cannot be a title.

Kindly consider and change the title into a statement and not a question, the question can be included as part of your introduction, along with your aim of study

4

Data absent

Kindly include the data to reveal instances about gender differences influencing health.

Please state clearly the impact (state wise, nationally and globally) and effect on people’s health, physically, socially, mentally e.t.c

Which is one of the reasons why you must have chosen the topic.

Study Rationale

Invariably including a strong rationale for this study would have brought out the best of this article

Thank you

7. PLOS authors have the option to publish the peer review history of their article (what does this mean?). If published, this will include your full peer review and any attached files.

Reviewer #1: No

Reviewer #2: No

Reviewer #3: **Yes: **Olaoluwa Pheabian Akinwale

Reviewer #4: No

Reviewer #5: No

---

## [Author Response · Author response to Decision Letter 1]

1 Dec 2021

Dear Editor, 

We are submitting the revised version of our manuscript “Are 2020’s medical students still suffering from gender stereotypes? An Italian cross-sectional study”, according to the comments of the reviewers.

Rev 1: 

1 Title issue Very good article but my concern is that a research question cannot be a title.

Kindly consider and change the title into a statement and not a question, the question can be included as part of your introduction, along with your aim of study

4 Data absent Kindly include the data to reveal instances about gender differences influencing health.

Please state clearly the impact (state wise, nationally and globally) and effect on people’s health, physically, socially, mentally e.t.c

Which is one of the reasons why you must have chosen the topic.

Study Rationale Invariably including a strong rationale for this study would have brought out the best of this article

We would like to thank you for your useful considerations.

We modified title: “Gender sensitivity and stereotypes in medical university students: An Italian cross-sectional study”.

We have also modified the introduction as follows:

“A Danish study with a sample of 6.9 million patients, reports that women receive a diagnosis of disease 4 years later than men(8). Another study reports that the in-hospital mortality of an acute myocardial infarction (AMI) is higher in women than in men up to 70 years of age and survival after 6 months of AMI is lower in women(9).”

“The focus on a medicine that takes gender differences into account begins in the 80s with the signing by the UN (United Nation) of a convention aimed at eliminating all forms of discrimination against women(22). In Europe, a growing interest in gender medicine led to the inclusion of this issue in the new Horizon 2020 research funding program(22). In Italy, a Plan for the application and diffusion of Gender Specific Medicine (2019) aims to spread the principles of gender specific medicine that are not yet fully and adequately implemented in medical academic training (22). For this reason, the aim of this study is to evaluate gender awareness through N-GAMS scale and its predictors in Italian male and female medical students, since no data exist in literature about gender awareness in our national context.”

Finally, we want to thank the other reviewers for their helpful consideration.

---

## [Decision Letter · Decision Letter 2]

23 Dec 2021

Gender sensitivity and stereotypes in medical university students: An Italian cross-sectional study

PONE-D-21-25375R2

Dear Dr. Boietti,

We’re pleased to inform you that your manuscript has been judged scientifically suitable for publication and will be formally accepted for publication once it meets all outstanding technical requirements.

Kind regards,

Ramune Jacobsen

Academic Editor

PLOS ONE

Reviewer's Responses to Questions

**Comments to the Author**

1. If the authors have adequately addressed your comments raised in a previous round of review and you feel that this manuscript is now acceptable for publication, you may indicate that here to bypass the “Comments to the Author” section, enter your conflict of interest statement in the “Confidential to Editor” section, and submit your "Accept" recommendation.

Reviewer #2: All comments have been addressed

Reviewer #3: All comments have been addressed

Reviewer #4: All comments have been addressed

Reviewer #5: All comments have been addressed

2. Is the manuscript technically sound, and do the data support the conclusions?

Reviewer #2: Yes

Reviewer #3: Yes

Reviewer #4: Yes

Reviewer #5: Yes

3. Has the statistical analysis been performed appropriately and rigorously? 

Reviewer #2: Yes

Reviewer #3: Yes

Reviewer #4: Yes

Reviewer #5: Yes

4. Have the authors made all data underlying the findings in their manuscript fully available?

Reviewer #2: Yes

Reviewer #3: Yes

Reviewer #4: Yes

Reviewer #5: Yes

5. Is the manuscript presented in an intelligible fashion and written in standard English?

Reviewer #2: Yes

Reviewer #3: Yes

Reviewer #4: Yes

Reviewer #5: Yes

6. Review Comments to the Author

Reviewer #2: All the comments and questions raised in previous versions of the manuscript have been addressed by the authors.

Reviewer #3: The manuscript was presented in an intelligible fashion and written in standard English, and the authors have addressed satisfactorily all my concerns in their responses.

Reviewer #4: (No Response)

Reviewer #5: All the best in the future

Pages Queries My suggestions to the author

1 Title issue Very good article but my concern is that a research question cannot be a title.

Kindly consider and change the title into a statement and not a question, the question can be included as part of your introduction, along with your aim of study

4 Data absent Kindly include the data to reveal instances about gender differences influencing health.

Please state clearly the impact (state wise, nationally and globally) and effect on people’s health, physically, socially, mentally e.t.c

Which is one of the reasons why you must have chosen the topic.

Study Rationale Invariably including a strong rationale for this study would have brought out the best of this article

7. PLOS authors have the option to publish the peer review history of their article (what does this mean?). If published, this will include your full peer review and any attached files.

Reviewer #2: No

Reviewer #3: **Yes: **Prof Olaoluwa Pheabian Akinwale

Reviewer #4: No

Reviewer #5: No

---

## [Editor Report · Acceptance letter]

28 Dec 2021

PONE-D-21-25375R2 

Gender sensitivity and stereotypes in medical university students: An Italian cross-sectional study 

Dear Dr. Boietti:

I'm pleased to inform you that your manuscript has been deemed suitable for publication in PLOS ONE. Congratulations! Your manuscript is now with our production department. 

Kind regards, 

on behalf of

Dr. Ramune Jacobsen 

Academic Editor

PLOS ONE